# Efficient Graph Generation with Graph Recurrent Attention Networks

**Renjie Liao**[1,2,3], **Yujia Li**[4], **Yang Song**[5], **Shenlong Wang**[1,2,3],
**William L. Hamilton**[6,7], **David Duvenaud**[1,3], **Raquel Urtasun**[1,2,3], **Richard Zemel**[1,3,8]
University of Toronto[1], Uber ATG Toronto[2], Vector Institute[3],
DeepMind[4], Stanford University[5], McGill University[6],
Mila – Quebec Artificial Intelligence Institute[7], Canadian Institute for Advanced Research[8]
{rjliao, slwang, duvenaud, urtasun, zemel}@cs.toronto.edu
{yujiali, charlienash}@google.com, yangsong@cs.stanford.edu, wlh@cs.mcgill.ca

## Abstract

We propose a new family of efficient and expressive deep generative models of graphs, called Graph Recurrent Attention Networks (GRANs). Our model generates graphs one block of nodes and associated edges at a time. The block size and sampling stride allow us to trade off sample quality for efficiency. Compared to previous RNN-based graph generative models, our framework better captures the auto-regressive conditioning between the already-generated and to-be-generated parts of the graph using Graph Neural Networks (GNNs) with attention. This not only reduces the dependency on node ordering but also bypasses the long-term bottleneck caused by the sequential nature of RNNs. Moreover, we parameterize the output distribution per block using a mixture of Bernoulli, which captures the correlations among generated edges within the block. Finally, we propose to handle node orderings in generation by marginalizing over a family of canonical orderings. On standard benchmarks, we achieve state-of-the-art time efficiency and sample quality compared to previous models. Additionally, we show our model is capable of generating large graphs of up to 5K nodes with good quality. Our code is released at: https://github.com/lrjconan/GRAN.

## 1 Introduction

Graphs are the natural data structure to represent relational and structural information in many domains, such as knowledge bases, social networks, molecule structures and even the structure of probabilistic models. The ability to generate graphs therefore has many applications; for example, a generative model of molecular graph structures can be employed for drug design [10, 21, 19, 36], generative models for computation graph structures can be useful in model architecture search [35], and graph generative models also play a significant role in network science [34, 1, 18].

The study of generative models for graphs dates back at least to the early work by Erdős and Rényi [8] in the 1960s. These traditional approaches to graph generation focus on various families of random graph models [38, 8, 13, 34, 2, 1], which typically formalize a simple stochastic generation process (e.g., random, preferential attachment) and have well-understood mathematical properties. However, due to their simplicity and hand-crafted nature, these random graph models generally have limited capacity to model complex dependencies and are only capable of modeling a few statistical properties of graphs. For example, Erdős–Rényi graphs do not have the heavy-tailed degree distribution that is typical for many real-world networks.

More recently, building graph generative models using neural networks has attracted increasing attention [27, 10, 21]. Compared to traditional random graph models, these deep generative models have a

greater capacity to learn structural information from data and can model graphs with complicated topology and constrained structural properties, such as molecules.

Several paradigms have been developed in the context of graph generative models. The first category of models generates the components of the graph independently or with only weak dependency structure of the generation decisions. Examples of these models include the variational auto-encoder (VAE) model with convolutional neural networks for sequentialized molecule graphs [10] and Graph VAE models [16, 29, 23]. These models generate the individual entries in the graph adjacency matrix (i.e., edges) independently given the latents; this independence assumption makes the models efficient and generally parallelizable but can seriously compromise the quality of the generated graphs [21, 37].

The second category of deep graph generative models make auto-regressive decisions when generating the graphs. By modeling graph generation as a sequential process, these approaches naturally accommodate complex dependencies between generated edges. Previous work on auto-regressive graph generation utilize recurrent neural networks (RNNs) on domain-specific sequentializations of graphs (e.g., SMILES strings) [27], as well as auto-regressive models that sequentially add nodes and edges [21, 37] or small graph motifs [14] (via junction-tree VAEs). However, a key challenge in this line of work is finding a way to exploit the graph structure during the generation process. For instance, while applying RNNs to SMILES strings (as in [27]) is computationally efficient, this approach is limited by the domain-specific sequentialization of the molecule graph structure. A similar issue arises in junction tree VAEs that rely on small molecule-specific graph motifs. On the other hand, previous work that exploits general graph structures using graph neural networks (GNNs) [21] do not scale well, with a maximum size of the generated graphs not exceeding 100 nodes.

Currently, the most scalable auto-regressive framework that is both general (i.e., not molecule-specific) and able to exploit graph structure is the GraphRNN model [37], where the entries in a graph adjacency matrix are generated sequentially, one entry or one column at a time through an RNN. Without using GNNs in the loop, these models can scale up significantly, to generate graphs with hundreds of nodes. However, the GraphRNN model has some important limitations: (1) the number of generation steps in the full GraphRNN is still very large ($O(N^2)$ for the best model, where $N$ is the number of nodes); (2) due to the sequential ordering, two nodes nearby in the graph could be far apart in the generation process of the RNN, which presents significant bottlenecks in handling such long-term dependencies. In addition, handling permutation invariance is vital for generative models on graphs, since computing the likelihood requires marginalizing out the possible permutations of the node orderings for the adjacency matrix. This becomes more challenging as graphs scale up, since it is impossible to enumerate all permutations as was done in previous methods for molecule graphs [21]. GraphRNN relies on the random breadth-first search (BFS) ordering of the nodes across all graphs, which is efficient to compute but arguably suboptimal.

In this paper, we propose an efficient auto-regressive graph generative model, called Graph Recurrent Attention Network (GRAN), which overcomes the shortcomings of previous approaches. In particular,

- Our approach is a generation process with $O(N)$ auto-regressive decision steps, where a block of nodes and associated edges are generated at each step, and varying the block size along with the sampling stride allow us to explore the efficiency-quality trade-off.

- We propose an attention-based GNN that better utilizes the topology of the already generated part of the graph to effectively model complex dependencies between this part and newly added nodes. GNN reduces the dependency on the node ordering as it is permutation equivariant w.r.t. the node representations. Moreover, the attention mechanism helps distinguish multiple newly added nodes.

- We parameterize the output distribution per generation step using a mixture of Bernoullis, which can capture the correlation between multiple generated edges.

- We propose a solution to handle node orderings by marginalizing over a family of "canonical" node orderings (e.g., DFS, BFS, or k-core). This formulation has a variational interpretation as adaptively choosing the optimal ordering for each graph.

Altogether, we obtain a model that achieves the state-of-the-art performance on standard benchmarks and permits flexible trade-off between generation efficiency and sample quality. Moreover, we successfully apply our model to a large graph dataset with graphs up to 5k nodes, the scale of which is significantly beyond the limits of existing deep graph generative models.

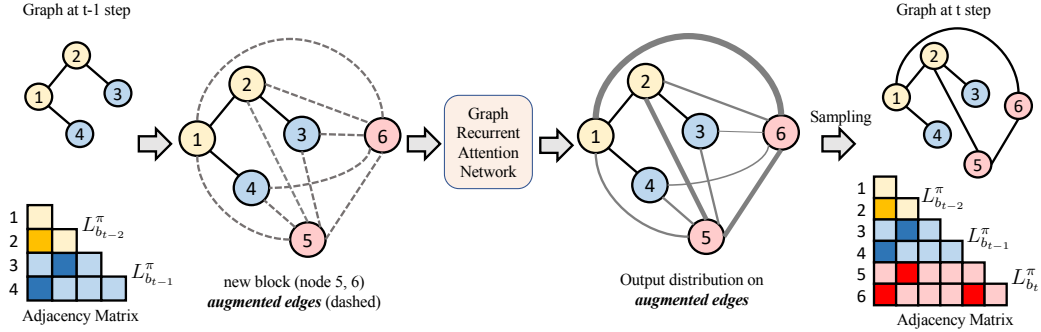

Figure 1: Overview of our model. Dashed lines are *augmented edges*. Nodes with the same color belong to the same block (block size $= 2$). In the middle right, for simplicity, we visualize the output distribution as a single Bernoulli where the line width indicates the probability of generating the edge.

## 2 Model

### 2.1 Representation of Graphs and the Generation Process

We aim to learn a distribution $p(G)$ over simple graphs that have at most one edge between any pair of nodes. Given a simple graph $G = (V, E)$ and an ordering $\pi$ over the nodes in the graph, we have a bijection $A^\pi \Leftrightarrow (G, \pi)$ between the set of possible adjacency matrices and the set of node-ordered graphs. We use the superscript $\pi$ in $A^\pi$ to emphasize that the adjacency matrix implicitly assumes a node ordering via the order of its rows/columns.

Based on the bijection above, we can model $p(G) = \sum_\pi p(G, \pi) = \sum_\pi p(A^\pi)$ by modeling the distribution $p(A^\pi)$ over adjacency matrices. For undirected graphs, $A^\pi$ is symmetric, thus we can model only the lower triangular part of $A^\pi$, denoted as $L^\pi$. Intuitively, our approach generates the entries in $L^\pi$ one row (or one block of rows) at a time, generating a sequence of vectors $\{L_i^\pi | i = 1, \cdots, |V|\}$ where $L_i^\pi \in \mathbb{R}^{1 \times |V|}$ is the $i$-th row of $L^\pi$ padded with zeros. The process continues until we reach the maximum number of time steps (rows). Once $L^\pi$ is generated, we have $A^\pi = L^\pi + L^{\pi\top}$.[1]

Note that our approach generates all the entries in one row (or one block of rows) in one pass conditioned on the already generated graph. This significantly shortens the sequence of auto-regressive graph generation decisions by a factor of $O(N)$, where $N = |V|$. By increasing the block size and the stride of generation, we trade-off model expressiveness for speed. We visualize the overall process per generation step in Figure 1.

### 2.2 Graph Recurrent Attention Networks

Formally, our model generates one block of $B$ rows of $L^\pi$ at a time. The $t$-th block contains rows with indices in $\boldsymbol{b_t} = \{B(t-1) + 1, ..., Bt\}$. We use bold subscripts to denote a block of rows or vectors, and normal subscripts for individual rows or vectors. The number of steps to generate a graph is therefore $T = \lceil N/B \rceil$. We factorize the probability of generating $L^\pi$ as

$$p(L^\pi) = \prod_{t=1}^{T} p(L_{\boldsymbol{b_t}}^\pi | L_{\boldsymbol{b_1}}^\pi, \cdots, L_{\boldsymbol{b_{t-1}}}^\pi). \tag{1}$$

The conditional distribution $p(L_{\boldsymbol{b_t}}^\pi | L_{\boldsymbol{b_1}}^\pi, \cdots, L_{\boldsymbol{b_{t-1}}}^\pi)$ defines the probability of generating the current block (all edges from the nodes in this block to each other and to the nodes generated earlier) conditioned on the existing graph. RNNs are standard neural network models for handling this type of sequential dependency structures. However, two nodes which are nearby in terms of graph topology may be far apart in the sequential node ordering. This causes the so called long-term bottleneck for RNNs To improve the conditioning, we propose to use GNNs rather than RNNs to make the

generation decisions of the current block directly depend on the graph structure. We do not carry hidden states of GNNs from one generation step to the next. By doing so, we enjoy the parallel training similar to PixelCNN [31], which is more efficient than the training method of typical deep auto-regressive models. We now explain the details of one generation step as below.

**Node Representation:** At the $t$-th generation step, we first compute the initial node representations of the already-generated graph via a linear mapping,

$$h^0_{\boldsymbol{b}_i} = WL^\pi_{\boldsymbol{b}_i} + b, \qquad \forall i < t. \tag{2}$$

A block $L^\pi_{\boldsymbol{b}_i}$ is represented as a vector $[L^\pi_{B(i-1)+1}, ..., L^\pi_{Bi}] \in \mathbb{R}^{BN}$, where $[.]$ is the vector concatenation operator and $N$ is the maximum allowed graph size for generation; graphs smaller than this size will have the $L^\pi_i$ vectors padded with 0s. These $h$ vectors will then be used as initial node representations in the GNN, hence a superscript of 0. For the current block $L^\pi_{\boldsymbol{b}_t}$, since we have not generated anything yet, we set $h^0_{\boldsymbol{b}_t} = \mathbf{0}$. Note also that $h_{\boldsymbol{b}_t} \in \mathbb{R}^{BH}$ contains a representation vector of size $H$ for each node in the block $\boldsymbol{b}_t$. In practice, computing $h^0_{\boldsymbol{b}_{t-1}}$ alone at the $t$-th generation step is enough as $\{h^0_{\boldsymbol{b}_i} | i < t - 1\}$ can be cached from previous steps. The main goal of this linear layer is to reduce the embedding size to better handle large scale graphs.

**Graph Neural Networks with Attentive Messages:** From these node representations, all the edges associated with the current block are generated using a GNN. These edges include connections within the block as well as edges linking the current block with the previously generated nodes.

For the $t$-th generation step, we construct a graph $G_t$ that contains the already-generated subgraph of $B(t-1)$ nodes and the edges between these nodes, as well as the $B$ nodes in the block to be generated. For these $B$ new nodes, we add edges to connect them with each other and the previous $B(t-1)$ nodes. We call these new edges *augmented edges* and depict them with dashed lines in Figure 1. We then use a GNN on this augmented graph $G_t$ to get updated node representations that encode the graph structure. More concretely, the $r$-th round of message passing in the GNN is implemented with the following equations:

$$m^r_{ij} = f(h^r_i - h^r_j), \tag{3} \qquad a^r_{ij} = \text{Sigmoid}\left(g(\tilde{h}^r_i - \tilde{h}^r_j)\right), \tag{5}$$

$$\tilde{h}^r_i = [h^r_i, x_i], \tag{4} \qquad h^{r+1}_i = \text{GRU}(h^r_i, \sum\nolimits_{j \in \mathcal{N}(i)} a^r_{ij} m^r_{ij}). \tag{6}$$

Here $h^r_i$ is the hidden representation for node $i$ after round $r$, and $m^r_{ij}$ is the message vector from node $i$ to $j$. $\tilde{h}^r_i$ is $h^r_i$ augmented with a $B$-dimensional binary mask $x_i$ indicating whether node $i$ is in the existing $B(t-1)$ nodes (in which case $x_i = \mathbf{0}$), or in the new block of $B$ nodes ($x_i$ is a one-of-$B$ encoding of the relative position of node $i$ in this block). $a^r_{ij}$ is an attention weight associated with edge $(i, j)$. The dependence on $\tilde{h}^r_i$ and $\tilde{h}^r_j$ makes it possible for the model to distinguish between existing nodes and nodes in the current block, and to learn different attention weights for different types of edges. Both the message function $f$ and the attention function $g$ are implemented as 2-layer MLPs with ReLU nonlinearities. Finally, the node representations are updated through a GRU similar to [20] after aggregating all the incoming messages through an attention-weighted sum over the neighborhood $\mathcal{N}(i)$ for each node $i$. Note that the node representations $h^0_{\boldsymbol{b}_{t-1}}$ from Eq. 2 are reused as inputs to the GNN for all subsequent generation steps $t' \geq t$. One can also untie the parameters of the model at each propagation round to improve the model capacity.

**Output Distribution:** After $R$ rounds of message passing, we obtain the final node representation vectors $h^R_i$ for each node $i$, and then model the probability of generating edges in the block $L^\pi_{\boldsymbol{b}_t}$ via a mixture of Bernoulli distributions:

$$p(L^\pi_{\boldsymbol{b}_t} | L^\pi_{\boldsymbol{b}_1}, ..., L^\pi_{\boldsymbol{b}_{t-1}}) = \sum_{k=1}^K \alpha_k \prod_{i \in \boldsymbol{b}_t} \prod_{1 \leq j \leq i} \theta_{k,i,j}, \tag{7}$$

$$\alpha_1, \ldots, \alpha_K = \text{Softmax}\left(\sum\nolimits_{i \in \boldsymbol{b}_t, 1 \leq j \leq i} \text{MLP}_\alpha(h^R_i - h^R_j)\right), \tag{8}$$

$$\theta_{1,i,j}, \ldots, \theta_{K,i,j} = \text{Sigmoid}\left(\text{MLP}_\theta(h^R_i - h^R_j)\right) \tag{9}$$

where both $\text{MLP}_\alpha$ and $\text{MLP}_\theta$ contain two hidden layers with ReLU nonlinearities and have $K$-dimensional outputs. Here $K$ is the number of mixture components. When $K = 1$, the distribution degenerates to Bernoulli which assumes the independence of each potential edge conditioned on the existing graph. This is a strong assumption and may compromise the model capacity. We illustrate the single Bernoulli output distribution in the middle right of Figure 1 using the line width. When $K > 1$, the generation of individual edges are not independent due to the latent mixture components. Therefore, the mixture model provides a cheap way to capture dependence in the output distribution, as within each mixture component the distribution is fully factorial, and all the mixture components can be computed in parallel.

**Block Size and Stride:**   The main limiting factor for graph generation speed is the number of generation steps $T$, as the $T$ steps have to be performed sequentially and therefore cannot benefit from parallelization. To improve speed, it is beneficial to use a large block size $B$, as the number of steps needed to generate graphs of size $N$ is $\lceil N/B \rceil$. On the other hand, as $B$ grows, modeling the generation of large blocks becomes increasingly difficult, and the model quality may suffer.

We propose "strided sampling" to allow a trained model to improve its performance without being retrained or fine-tuned. More concretely, after generating a block of size $B$, we may choose to only keep the first $S$ rows in the block, and in the next step generate another block starting from the $(S + 1)$-th row. We call $S$ $(1 \leq S \leq B)$ the "stride" for generation, inspired by the stride in convolutional neural networks. The standard generation process corresponds to a stride of $S = B$. With $S < B$, neighboring blocks have an overlap of $B - S$ rows, and $T = \lfloor (N - B)/S \rfloor + 1$ steps is needed to generate a graph of size $N$. During training, we train with block size of $B$ and stride size of 1, hence learning to generate the next $B$ rows conditioned on all possible subgraphs under a particular node ordering; while at test time we can use the model with different stride values. Setting $S$ to $B$ maximizes speed, and using smaller $S$ can improve quality, as the dependency between the rows in a block can be modeled by more than 1 steps.

## 2.3   Learning with Families of Canonical Orderings

Ordering is important for an auto-regressive model. Previous work [21, 37] explored learning and generation under a canonical ordering (e.g., based on BFS), and Li et al. [21] also explored training with a uniform random ordering. Here, we propose to train under a chosen family of canonical orderings, allowing the model to consider multiple orderings with different structural biases while avoiding the intractable full space of factorially-many orderings. Similar strategy has been exploited for learning relational pooling function in [25]. Concretely, we aim to maximize the log-likelihood $\log p(G) = \log \sum_\pi p(G, \pi)$. However this computation is intractable due to the number of orderings $\pi$ being factorial in the graph size. We therefore limit the set of orderings to a family of canonical orderings $\mathcal{Q} = \{\pi_1, ..., \pi_M\}$, and learn to maximize a lower bound

$$\log p(G) \geq \log \sum_{\pi \in \mathcal{Q}} p(G, \pi) \tag{10}$$

instead. Since $\mathcal{Q}$ is a strict subset of the $N!$ orderings, $\log \sum_{\pi \in \mathcal{Q}} p(G, \pi)$ is a valid lower bound of the true log-likelihood, and it is a tighter bound than any single term $\log p(G, \pi)$, which is the objective for picking a single arbitrary ordering and maximizing the log-likelihood under that ordering, a popular training strategy [37, 21]. On the other hand, increasing the size of $\mathcal{Q}$ can make the bound tighter. Choosing a set $\mathcal{Q}$ of the right size can therefore achieve a good trade-off between tightness of the bound (which usually correlates with better model quality) and computational cost.

**Variational Interpretation:**   This new objective has an intuitive variational interpretation. To see this, we write out the variational evidence lower bound (ELBO) on the log-likelihood,

$$\log p(G) \geq \mathbb{E}_{q(\pi|G)}[\log p(G, \pi)] + \mathcal{H}(q(\pi|G)), \tag{11}$$

where $q(\pi|G)$ is a variational posterior over orderings given the graph $G$. When restricting $\pi$ to a set of $M$ canonical distributions, $q(\pi|G)$ is simply a categorical distribution over $M$ items, and the optimal $q^*(\pi|G)$ can be solved analytically with Lagrange multipliers, as

$$q^*(\pi|G) = p(G, \pi) / \left( \sum_{\pi \in \mathcal{Q}} p(G, \pi) \right) \qquad \forall \pi \in \mathcal{Q}. \tag{12}$$

Substituting $q(\pi|G)$ into Eq. 11, we get back to the objective defined in Eq. 10. In other words, by optimizing the objective in Eq. 10, we are implicitly picking the optimal (combination of) node orderings from the set $\mathcal{Q}$ and maximizing the log-likelihood of the graph under this optimal ordering.

**Canonical Node Orderings:** Different types or domains of graphs may favor different node orderings. For example, some canonical orderings are more effective compared to others in molecule generation, as shown in [21]. Therefore, incorporating prior knowledge in designing $\mathcal{Q}$ would help in domain-specific applications. Since we aim at a universal deep generative model of graphs, we consider the following canonical node orderings which are solely based on graph properties: the default ordering used in the data[2], the node degree descending ordering, ordering of the BFS/DFS tree rooted at the largest degree node (similar to [37]), and a novel core descending ordering. We present the details of the core node ordering in the appendix. In our experiments, we explore different combinations of orderings from this generic set to form our $\mathcal{Q}$.

## 3 Related Work

In addition to the graph generation approaches mentioned in Sec. 1, there are a number of other notable works that our research builds upon.

**Traditional approaches**. Exponential random graphs model (ERGMs) [33] are an early approach to learning a generative graph models from data. ERGMs rely on an expressive probabilistic model that learns weights over node features to model edge likelihoods, but in practice, this approach is limited by the fact that it only captures a set of hand-engineered graph-sufficient statistics. The Kronecker graph model [18] relies on Kronecker matrix products to efficiently generate large adjacency matrices. While scalable and able to learn some graph properties (e.g., degree distributions) from data, this approach remains highly-constrained in terms of the graph structures that it can represent.

**Non-autoregressive deep models**. A number of approaches have been proposed to improve non-auto-regressive deep graph generative models. For example, Grover et al. [11] use a graph neural network encoder and an iterative decoding procedure to define a generative model over a single fixed set of nodes. Ma et al. [24], on the other hand, propose to regularize the graph VAE objective with domain-specific constraints; however, as with other previous work on non-autoregressive, graph VAE approaches, they are only able to scale to graphs with less than one hundred nodes. NetGAN [4] builds a generative adversarial network on top of random walks over the graph. A sampled graph is then constructed based on multiple generated random walks. Relying on the reversible GNNs and graph attention networks [32], Liu et al. [22] propose a normalizing flow based prior within the framework of graphVAE. However, the graph in the latent space is assumed to be fully connected which significantly limits the scalability.

**Auto-regressive deep models**. For auto-regressive models, Dai et al. [6] chose to model the sequential graphs using RNNs, and enforced constraints in the output space using attribute grammars to ensure the syntactic and semantic validity. Similarly, Kusner et al. [17] predicted the logits of different graph components independently but used syntactic constraints that depend on context to guarantee validity. Jin et al. [14] proposed a different approach to molecule graph generation by converting molecule graphs into equivalent junction trees. This approach works well for molecules but is not efficient for modeling large tree-width graphs. The molecule generation model of Li et al. [19] is the most similar to ours, where one column of the adjacency matrix is also generated at each step, and the generation process is modeled as either Markovian or recurrent using a global RNN, which works well for molecule generation. Chu et al. [5] improve GraphRNN with a random-walk based encoder and successfully apply the model to road layout generation. Unlike these previous works, which focus on generating domain-specific and relatively small graphs, *e.g.*, molecules, we target the problem of efficiently generating large and generic graphs.

## 4 Experiments

In this section we empirically verify the effectiveness of our model on both synthetic and real graph datasets with drastically varying sizes and characteristics.

## 4.1 Dataset and Evaluation Metrics

Our experiment setup closely follows You et al. [37]. To benchmark our GRAN against the models proposed in the literature, we utilize one synthetic dataset containing random grid graphs and two real world datasets containing protein and 3D point clouds respectively.

**Datasets:** (1) Grid: We generate 100 standard 2D grid graphs with $100 \leq |V| \leq 400$. (2) Protein: This dataset contains 918 protein graphs [7] with $100 \leq |V| \leq 500$. Each protein is represented by a graph, where nodes are amino acids and two nodes are connected by an edge if they are less than 6 Angstroms away. (3) Point Cloud: FirstMM-DB is a dataset of 41 simulated 3D point clouds of household objects [26] with an average graph size of over 1k nodes, and maximum graph size over 5k nodes. Each object is represented by a graph where nodes represent points. Edges are connected for k-nearest neighbors which are measured w.r.t. Euclidean distance of the points in 3D space. We use the same protocol as [37] and create random $80\%$ and $20\%$ splits of the graphs in each dataset for training and testing. $20\%$ of the training data in each split is used as the validation set. We generate the same amount of samples as the test set for each dataset.

**Metrics:** Evaluating generative models is known to be challenging [30]. Since it is difficult to measure likelihoods for any auto-regressive graph generative model that relies on an ordering, we follow [37, 21] and evaluate model performance by comparing the distributions of graph statistics between the generated and ground truth graphs. In previous work, You et al. [37] computed degree distributions, clustering coefficient distributions, and the number of occurrence of all orbits with 4 nodes, and then used the maximum mean discrepancy (MMD) over these graph statistics, relying on Gaussian kernels with the first Wasserstein distance, *i.e.*, earth mover's distance (EMD), in the MMD. In practice, we found computing this MMD with the Gaussian EMD kernel to be very slow for moderately large graphs. Therefore, we use the total variation (TV) distance, which greatly speeds up the evaluation and is still consistent with EMD. In addition to the node degree, clustering coefficient and orbit counts (used by [36]), we also compare the spectra of the graphs by computing the eigenvalues of the normalized graph Laplacian (quantized to approximate a probability density). This spectral comparison provides a view of the global graph properties, whereas the previous metrics focus on local graph statistics.

## 4.2 Benchmarking Sample Quality

In the first experiment we compare the quality of our GRAN model against other existing models in the literature including GraphVAE [29], GraphRNN and its variant GraphRNN-S [37]. We also add a Erdos-Renyi baseline of which the edge probability is estimated via maximum likelihood over the training graphs. For a fair comparison, we control the data split to be exactly same for all methods. We implement a GraphVAE model where the encoder is a 3-layer GCN and decoder is a MLP with 2 hidden layers. All hidden dimensions are set to 128. For GraphRNN and GraphRNN-S, we use the best setting reported in the paper and re-trained the model with our split. We also tried to run DeepGMG model [21] but failed to obtain results in a reasonable amount of time due to its scalability issue on these datasets. For our GRAN, hidden dimensions are set to 128, 512 and 256 on three datasets respectively. Block size and stride are both set to 1. The number of Bernoulli mixtures is 20 for all experiments. We stack 7 layers of GNNs and unroll each layer for 1 step. All of our models are trained with Adam optimizer [15] and constant learning rate 1e-4. We choose the best model based on the validation set. The sample evaluation results on the test set are reported in Table 1, with a few sample graphs generated by the models shown in Figure 2. For all the metrics, smaller is better.

Note that none of the Graph-VAE, GraphRNN and GraphRNN-S models were able to scale to the point cloud dataset due to out-of-memory issues, and the running time of GraphRNN models becomes prohibitively long for large graphs. Overall, our proposed GRAN model achieves the state-of-the-art performance on all benchmarks in terms of sample quality. On the other hand, from Figure 2, we can see that even though the quantitative metric of GraphRNN is similar to ours, the visual difference of the generated grid graphs is particularly noticeable. This implies that the current set of graph statistics may not give us a complete picture of model performance. We show more visual examples and results on one more synthetic random lobster graph dataset in the appendix.

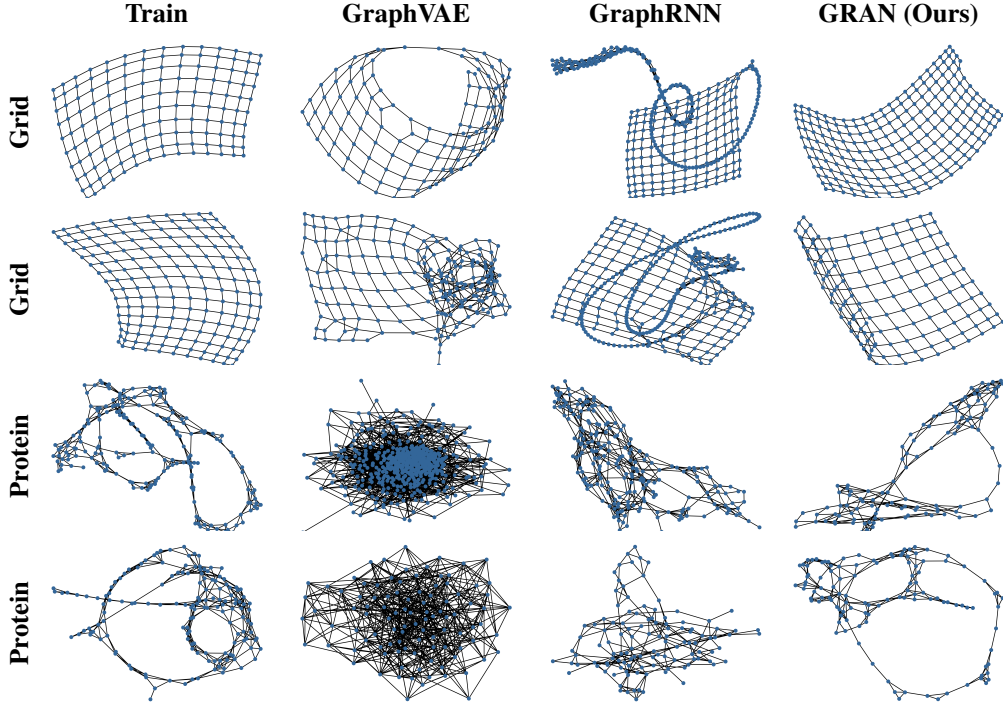

| | Train | GraphVAE | GraphRNN | GRAN (Ours) |
|---|---|---|---|---|
| Grid | | | | |
| Grid | | | | |
| Protein | | | | |
| Protein | | | | |

Figure 2: Visualization of sample graphs generated by different models.

| | Grid | | | | Protein | | | | 3D Point Cloud | | | |
|---|---|---|---|---|---|---|---|---|---|---|---|---|
| | $\|V\|_{max} = 361, \|E\|_{max} = 684$<br>$\|V\|_{avg} \approx 210, \|E\|_{avg} \approx 392$ | | | | $\|V\|_{max} = 500, \|E\|_{max} = 1575$<br>$\|V\|_{avg} \approx 258, \|E\|_{avg} \approx 646$ | | | | $\|V\|_{max} = 5037, \|E\|_{max} = 10886$<br>$\|V\|_{avg} \approx 1377, \|E\|_{avg} \approx 3074$ | | | |
| | Deg. | Clus. | Orbit | Spec. | Deg. | Clus. | Orbit | Spec. | Deg. | Clus. | Orbit | Spec. |
| Erdos-Renyi | 0.79 | 2.00 | 1.08 | 0.68 | $5.64e^{-2}$ | 1.00 | 1.54 | $9.13e^{-2}$ | 0.31 | 1.22 | 1.27 | $4.26e^{-2}$ |
| GraphVAE* | $7.07e^{-2}$ | $7.33e^{-2}$ | 0.12 | $1.44e^{-2}$ | 0.48 | $7.14e^{-2}$ | 0.74 | 0.11 | - | - | - | - |
| GraphRNN-S | 0.13 | $3.73e^{-2}$ | 0.18 | 0.19 | $4.02e^{-2}$ | $\mathbf{4.79e^{-2}}$ | 0.23 | 0.21 | - | - | - | - |
| GraphRNN | $1.12e^{-2}$ | $\mathbf{7.73e^{-5}}$ | $\mathbf{1.03e^{-3}}$ | $\mathbf{1.18e^{-2}}$ | $1.06e^{-2}$ | 0.14 | 0.88 | $1.88e^{-2}$ | - | - | - | - |
| GRAN | $\mathbf{8.23e^{-4}}$ | $3.79e^{-3}$ | $1.59e^{-3}$ | $1.62e^{-2}$ | $\mathbf{1.98e^{-3}}$ | $4.86e^{-2}$ | $\mathbf{0.13}$ | $\mathbf{5.13e^{-3}}$ | $\mathbf{1.75e^{-2}}$ | $\mathbf{0.51}$ | $\mathbf{0.21}$ | $\mathbf{7.45e^{-3}}$ |

Table 1: Comparison with other graph generative models. For all MMD metrics, the smaller the better. *: our own implementation, -: not applicable due to memory issue, Deg.: degree distribution, Clus.: clustering coefficients, Orbit: the number of 4-node orbits, Spec.: spectrum of graph Laplacian.

## 4.3 Efficiency vs. Sample Quality

Another important feature of GRAN is its efficiency. In this experiment, we quantify the graph generation efficiency and show the efficiency-quality trade-off by varying the generation stride. The main results are reported in Figure 3, where the models are trained with block size 16. We trained our models on grid graphs and evaluated model quality on the validation set. To measure the run time for each setting we used a single GTX 1080Ti GPU. We measure the speed improvement by computing the ratio of GraphRNN average time per graph to that of ours. GraphRNN takes around 9.5 seconds to generate one grid graph on average. Our best performing model with stride 1 is about 6 times as fast as GraphRNN, increasing the sampling stride trades-off quality for speed. With a stride of 16, our model is more than 80x faster than GraphRNN, but the model quality is also noticeably worse. We leave the full details of quantitative results in the appendix.

## 4.4 Ablation Study

In this experiment we isolate the contributions of different parts of our model and present an ablation study. We again trained all models on the random grid graphs and report results on the validation set. From Table 2, we can see that increasing the number of Bernoulli mixtures improves the performance, especially w.r.t. the orbit metric. Note that since grid graphs are somewhat simple in terms of the

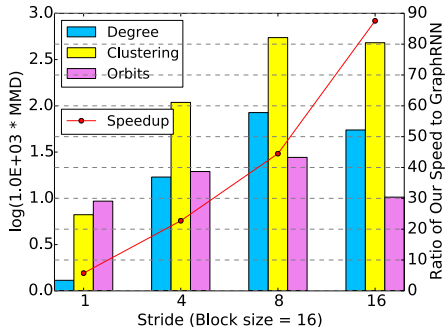

Figure 3: Efficiency vs. sample quality. The bar and line plots are the MMD (left y-axis) and speed ratio (right y-axis) respectively.

| B | K | $\mathcal{Q}$ | Deg. | Clus. | Orbit |
|---|---|---|---|---|---|
| 1 | 1 | $\{\pi_1\}$ | $1.51e^{-5}$ | 0 | $2.66e^{-5}$ |
| 1 | 20 | $\{\pi_1\}$ | $1.54e^{-5}$ | 0 | $4.27e^{-6}$ |
| 1 | 50 | $\{\pi_1\}$ | $1.70e^{-5}$ | 0 | $9.56e^{-7}$ |
| 1 | 20 | $\{\pi_1,\pi_2\}$ | $6.00e^{-2}$ | 0.16 | $2.48e^{-2}$ |
| 1 | 20 | $\{\pi_1,\pi_2,\pi_3\}$ | $8.99e^{-3}$ | $7.37e^{-3}$ | $1.69e^{-2}$ |
| 1 | 20 | $\{\pi_1,\pi_2,\pi_3,\pi_4\}$ | $2.34e^{-2}$ | $5.95e^{-2}$ | $5.21e^{-2}$ |
| 1 | 20 | $\{\pi_1,\pi_2,\pi_3,\pi_4,\pi_5\}$ | $4.11e^{-4}$ | $9.43e^{-3}$ | $6.66e^{-4}$ |
| 4 | 20 | $\{\pi_1\}$ | $1.69e^{-4}$ | 0 | $5.04e^{-4}$ |
| 8 | 20 | $\{\pi_1\}$ | $7.01e^{-5}$ | $4.89e^{-5}$ | $8.57e^{-5}$ |
| 16 | 20 | $\{\pi_1\}$ | $1.30e^{-3}$ | $6.65e^{-3}$ | $9.32e^{-3}$ |

Table 2: Ablation study on grid graphs. $B$: block size, $K$: number of Bernoulli mixtures, $\pi_1$: DFS, $\pi_2$: BFS, $\pi_3$: k-core, $\pi_4$: degree descent, $\pi_5$: default.

topology, the clustering coefficients and degree distribution are not discriminative enough to reflect the changes. We also tried different mixtures on protein graphs and confirmed that increasing the number of Bernoulli mixtures does bring significant gain of performance. We set it to 20 for all experiments as it achieves a good balance between performance and computational cost. We also compare with different families of canonical node orderings. We found that using all orderings and DFS ordering alone are similarly good on grid graphs. Therefore, we choose to use DFS ordering due to its lower computational cost. In general, the optimal set of canonical orderings $\mathcal{Q}$ is application/dataset dependent. For example, for molecule generation, adding SMILE string ordering would boost the performance of using DFS alone. More importantly, our learning scheme is simple yet flexible to support different choices of $\mathcal{Q}$. Finally, we test different block sizes while fixing the stride as $1$. It is clear that the larger the block size, the worse the performance, which indicates the learning task becomes more difficult.

## 5    Conclusion

In this paper, we propose the Graph Recurrent Attention Network (GRAN) for efficient graph generation. Our model generates one block of the adjacency matrix at a time through an $O(N)$ step generation process. The model uses GNNs with attention to condition the generation of a block on the existing graph, and we further propose a mixture of Bernoulli output distribution to capture correlations among generated edges per step. Varying the block size and sampling stride permits effective exploration of the efficiency-quality trade-off. We achieve state-of-the-art performance on standard benchmarks and show impressive results on a large graph dataset of which the scale is beyond the limit of any other deep graph generative models. In the future, we would like to explore this model in applications where graphs are latent or partially observed.

**Acknowledgments**

RL was supported by Connaught International Scholarship and RBC Fellowship. WLH was supported by a Canada CIFAR Chair in AI. RL, RU and RZ were supported in part by the Intelligence Advanced Research Projects Activity (IARPA) via Department of Interior/Interior Business Center (DoI/IBC) contract number D16PC00003. The U.S. Government is authorized to reproduce and distribute reprints for Governmental purposes notwithstanding any copyright annotation thereon. Disclaimer: the views and conclusions contained herein are those of the authors and should not be interpreted as necessarily representing the official policies or endorsements, either expressed or implied, of IARPA, DoI/IBC, or the U.S. Government.

## Footnotes

[1]One can easily generalize the aforementioned representation to directed graphs by augmenting one more pass to sequentially generate the upper triangular part.

[2]In our case, it is the default ordering used by NetworkX [12]

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
