[Supplementary Material]

# 6 Appendix

## 6.1 K-Core Node Ordering

The $k$-core graph decomposition has been shown to be very useful for modeling cohesive groups in social networks [28]. The $k$-core of a graph $G$ is a maximal subgraph that contains nodes of degree $k$ or more. Cores are nested, *i.e.*, $i$-core belongs to $j$-core if $i > j$, but they are not necessarily connected subgraphs. Most importantly, the core decomposition, *i.e.*, all cores ranked based on their orders, can be found in linear time (w.r.t. $|E|$) [3]. Based on the largest core number per node, we can uniquely determine a partition of all nodes, *i.e.*, disjoint sets of nodes which share the same largest core number. We then assign the core number of each disjoint set by the largest core number of its nodes. Starting from the set with the largest core number, we rank all nodes within the set in node degree descending order. Then we move to the second largest core and so on to obtain the final ordering of all nodes. We call this core descending ordering as *k-core node ordering*.

## 6.2 Lobster Graphs

We also compare our GRAN with other methods on another synthetic dataset, *i.e.*, random lobster graphs. A lobster graph [9] is a tree where each node is at most 2 hops away from a backbone path. Based on this property, one can easily verify whether a given graph is a lobster graph. Therefore, we add another evaluation metric, *i.e.*, the accuracy of the generated graph being a lobster graph. We generate 100 random lobster graphs [9] with $10 \leq |V| \leq 100$. We adopt the same training and testing protocol and report the results in Table 3. From the table, we can see that our GRAN performs slightly worse compared to GraphRNN on this small-size dataset. Potential reasons could be that (1) the size of the graph is small such that the long-term bottleneck problem is not significant for GraphRNN; (2) Our model neither carries over hidden states between consecutive generation steps like GraphRNN and nor exploits other memory mechanism. Therefore, it may be hard for our model to capture the global property of the graph which affects the accuracy metric.

Figure 4: Visualization of sample graphs generated by different models.

|  | Lobster | | | | |
|  | $|V|_{\max} = 100, |E|_{\max} = 99$ | | | | |
|  | Deg. | Clus. | Orbit | Spec. | Acc. |
| --- | --- | --- | --- | --- | --- |
| GraphVAE* | $2.09e^{-2}$ | $7.97e^{-2}$ | $1.43e^{-2}$ | $3.94e^{-2}$ | 0.09 |
| GraphRNN | $\mathbf{9.26e^{-5}}$ | **0.0** | $\mathbf{2.19e^{-5}}$ | $\mathbf{1.14e^{-2}}$ | **1.00** |
| GRAN | $3.73e^{-2}$ | **0.0** | $7.67e^{-4}$ | $2.71e^{-2}$ | 0.88 |

Table 3: Comparison with other deep graph generative models on random lobster graphs. * means our own implementation.

## 6.3 Full Ablation Study & Visual Examples

We provide the full details of the ablation study in Table 4. We also show more visual examples of generated graphs in Figure 5, Figure 4, and Figure 6.

| Block Size | Sample Stride | # Bernoulli Mixtures | $\mathcal{Q}$ | Deg. | Clus. | Orbit | Spec. | Run Time(s) |
|---|---|---|---|---|---|---|---|---|
| 1 | 1 | 1 | $\{\pi_1\}$ | $1.51e^{-5}$ | 0 | $2.66e^{-5}$ | $1.57e^{-2}$ | - |
| 1 | 1 | 20 | $\{\pi_1\}$ | $1.54e^{-5}$ | 0 | $4.27e^{-6}$ | $1.32e^{-2}$ | - |
| 1 | 1 | 50 | $\{\pi_1\}$ | $1.70e^{-5}$ | 0 | $9.56e^{-7}$ | $1.35e^{-2}$ | - |
| 1 | 1 | 20 | $\{\pi_2\}$ | 0.16 | 0.29 | 0.16 | 0.15 | - |
| 1 | 1 | 20 | $\{\pi_3\}$ | $1.43e^{-2}$ | $3.38e^{-3}$ | $1.34e^{-2}$ | $3.97e^{-2}$ | - |
| 1 | 1 | 20 | $\{\pi_4\}$ | $2.50e^{-3}$ | $1.46e^{-2}$ | $5.23e^{-3}$ | $1.64e^{-2}$ | - |
| 1 | 1 | 20 | $\{\pi_5\}$ | $7.12e^{-2}$ | $1.10e^{-3}$ | $7.81e^{-2}$ | $8.43e^{-2}$ | - |
| 1 | 1 | 20 | $\{\pi_1, \pi_2\}$ | $6.00e^{-2}$ | 0.16 | $2.48e^{-2}$ | $4.79e^{-2}$ | - |
| 1 | 1 | 20 | $\{\pi_1, \pi_2, \pi_3\}$ | $8.99e^{-3}$ | $7.37e^{-3}$ | $1.69e^{-2}$ | $2.84e^{-2}$ | - |
| 1 | 1 | 20 | $\{\pi_1, \pi_2, \pi_3, \pi_4\}$ | $2.34e^{-2}$ | $5.95e^{-3}$ | $5.21e^{-2}$ | $4.49e^{-2}$ | - |
| 1 | 1 | 20 | $\{\pi_1, \pi_2, \pi_3, \pi_4, \pi_5\}$ | $4.11e^{-4}$ | $9.43e^{-3}$ | $6.66e^{-4}$ | $1.57e^{-2}$ | - |
| 4 | 1 | 20 | $\{\pi_1\}$ | $1.69e^{-4}$ | 0 | $5.04e^{-4}$ | $2.02e^{-2}$ | - |
| 8 | 1 | 20 | $\{\pi_1\}$ | $7.01e^{-5}$ | $4.89e^{-5}$ | $8.57e^{-5}$ | $2.20e^{-2}$ | - |
| 16 | 1 | 20 | $\{\pi_1\}$ | $1.30e^{-3}$ | $6.65e^{-3}$ | $9.32e^{-3}$ | $2.01e^{-2}$ | 33.1 |
| 16 | 4 | 20 | $\{\pi_1\}$ | $1.70e^{-2}$ | 0.11 | $1.95e^{-2}$ | $2.39e^{-2}$ | 8.4 |
| 16 | 8 | 20 | $\{\pi_1\}$ | $8.45e^{-2}$ | 0.55 | $2.77e^{-2}$ | $2.61e^{-2}$ | 4.3 |
| 16 | 16 | 20 | $\{\pi_1\}$ | $5.48e^{-2}$ | 0.48 | $1.03e^{-2}$ | $2.50e^{-2}$ | 2.2 |

Table 4: Full results of ablation study on grid graphs. $\pi_1$: DFS, $\pi_2$: BFS, $\pi_3$: k-core, $\pi_4$: degree descent, $\pi_5$: default. The run time is measured by generating a minibatch of 20 graphs.

Figure 5: Visualization of sample graphs generated by different models.

Figure 6: Visualization of sample graphs generated by our model on the 3D point cloud dataset.