[Reviews · NeurIPS 2019]

Reviewer 1



This manuscript describes a new deep learning model called Graph Recurrent Attention Network (GRAN) for the generation of a generic graph. This model has the following unique features:1) combine a RNN-based generator with an attention-based GNN that can model complex, long-range dependencies; 2) only need O(N) steps where N is the graph size (i.e, #nodes). Each step may generate a block of nodes; 3) the algorithm can generate different graphs by different orderings of nodes. The authors have shown that their algorithm can generate graphs of similar quality but much more efficiently. Their experimental results also show that their algorithm indeed has some advantage over existing ones. One concern is about the presentation of the experimental results. The authors used a few metrics to measure the quality of the generated graphs, but did not explain how these metrics are calculated and how they shall be interpreted. For example, when two methods have the spectral score 0.01 and 0.02, respectively, how different are they?

Reviewer 2



Significance: 1. The paper is very well written and easy to follow. However, most of the experimental details are missing and in the absence of code makes it difficult to imagine. 2. For example, no where it is mentioned how Eq. (11) is trained with ? It says 2-layer GCN but what is the input node representations ? And what is the GT in order to train for mixture network ? 3. Current drawback is that the model need to know the maximum allowed graph size 'N' (required for RNN module) 4. The results in Table 1 suggests that GRAN (this approach) performs best for Grid data. On other two dataset GraphRNN is better. Clarification: 1. a's in Eq.(5) are scalars ? Are they normalized using sigmoid activation function ? 2. What happens if RNN modules for node generation are not used ? h can be initialized randomly and updated based on GRU outputs of GNN module or something similar. Please inclkude this in Table 2 of ablation study. 3. How is overall training performed ? Is it teacher forcing approach ? 4. Please include model size comparison.

Reviewer 3



The general presentation of the paper is quite good. The paper is well written, and the authors present convincing derivations of the statistical foundations of their approach. Key arguments are widely supported by mathematical facts or well placed references. The experimental evaluation presents convincing results, with both quantitative and visual content to assess the quality of the generated samples. However, the fact that most of the comparison is made on purely synthetic data is slightly disappointing. Indeed, the only real dataset used is the protein dataset. Yet, it only contains small graphs, which is not the configuration where the proposed method is supposed to be the most relevant. The authors could have made use of open source social interaction graphs datasets, so as to prove their point about scalability, and demonstrate high sample quality on real-world large graph datasets.

Reviewer 4



Here is an update of my review. I am keeping my score but have a concern. <> Some of your more recent experiments show that models without an RNN can perform well in some instances. But from my reading, the RNN was an important part of the message of your paper. You are finding a balance between independent generation (graph VAE) and introducing structural dependence with an RNN. Please be careful to make sure the message of the entire paper is clear, consistent, and as unified as possible. Perhaps you can offer some guidance to the reader when an RNN will be useful, and why it may not have been needed in some of your tasks. I know you mentioned SMILES. <> Optimization of the GNN. My question was not about the type and tuning of the optimizer (Adam, learning rate, etc). Sorry for not making it clear. My suggestion was to clearly indicate what the final and overall loss function is that you use to train all the weights. You have a GNN that defines a Bernoulli distribution over edges and another that learns a variational posterior over orderings, as well as an RNN. Overall, all these components build up a likelihood function that serves as the loss that can be used to _optimize_ the weights. In my humble opinion the final objective function could be more salient, especially for purposes of implementation in the reader's favorite framework. For instance, an Algorithm environment with a line that refers to the equations that altogether define the loss. However I know space can constrain that suggestion. ====================== This paper focuses on the approach of using an auto-regressive scheme for graph generative models which better captures dependencies among vertices in the graph. The authors note that many existing works leverage domain-specific knowledge rather than learning from the data and scale poorly. A central baseline that addresses these issues is GraphRNN which still suffers from O(N^2) generation steps (N is the number of vertices), may lose long-range dependency between distant vertices in the graph, and only presents one scheme for ordering vertices. The authors propose generating a graph by focusing on B rows in the lower-triangle of the adjacency matrix at a time (a "block" of size B). To generate a new block of the graph, rows of the already-generated portion of the lower triangle are passed to a GRU. The representations from the GRU are passed to a graph neural network with attention, and the resulting vertex representations can be used to calculate a probability of edge formation (via an MLP). To learn an ordering, the authors take a variational approach with a categorical distribution over orderings as the variational posterior. Notably, this generative scheme, dubbed GRAN, can be seen to address each of the concerns discussed previously. The experiments compare summary statistics of generated graphs to actual graphs in real and simulated data against the baselines of GraphVAE and GraphRNN, demonstrate a speed-quality tradeoff controlled by B, and perform an ablation study. * Pros* - The characterization of existing work and discussion of limitations are clear. - The stated goals are clear and the scheme developed is clearly motivated by these goals. - This is an important line of work and I am not aware of work that takes the proposed approach. - The authors propose an interesting approach for learning long-range dependencies and mixing over canonical orderings. - The experiments demonstrate that GRAN makes graph generation via an autoregressive approach more computationally efficient if a user is willing to compromise on quality. *Cons* Issues with the experimental design preclude me from rating this paper above a 6. A few details in the description of the algorithm/training are also lacking. In particular: - It appears that one train-test split was chosen to be consistent with a previous experiment. It has been discussed that this practice should be avoided, e.g. “Pitfalls of Graph Neural Network Evaluation” - The experiments do not report confidence intervals or any quantification of the variability in the results (related to above). - The choice/optimization of weights for the GNN that produces edge probabilities needs to be more clearly discussed. - If possible, experiments could address whether the proposed model better learns long-range dependencies. - It could be made clearer how the size of the generated graph is determined.

[Author Response · NeurIPS 2019]

We thank all the reviewers for their constructive comments. We first explain the update of our model which resolves
some of R2's questions. Then we address the common concern on the large scale graph generation followed by
individual responses. We will release our PyTorch implementation upon acceptance.

**Model Simplification & Extension**: After submission, we tried removing the RNN, thus simplifying our model to a
single GNN. With more layers and slightly more training epochs, a single GNN performs similarly to the model with
an RNN. Moreover, we extended the output distribution from a single Bernoulli distribution (Eq.7 in the paper) to a
mixture of Bernoullis, which can further model the dependency between the entries in a block without sacrificing the
speed, and achieves state-of-the-art performance on most metrics of Protein dataset ($c.f.$ Table 1).

| Protein Dataset | Deg. | Clus. | Orbit | Spec. | | SBM Graphs | Deg. | Clus. | Orbit |
|---|---|---|---|---|---|---|---|---|---|
| GraphVAE | $6.12e^{-2}$ | 0.17 | 0.82 | $9.28e^{-2}$ | | $G(5000, 0.05)$ | 0.78 | 1.93 | 0.37 |
| GraphRNN | $\mathbf{5.51}e^{-3}$ | 0.15 | 0.40 | $2.56e^{-2}$ | | $G(5000, 0.1)$ | 0.78 | 1.83 | 0.37 |
| GRAN-50 | $1.08e^{-2}$ | **0.08** | **0.30** | $\mathbf{8.97}e^{-3}$ | | GRAN-20 | **0.54** | **0.99** | **0.34** |

Table 1: GRAN-X: our model (RNN removed) with a mixture of X Bernoulli distributions. $G(n, p)$: Erdos Renyi
model with number of nodes $n$ and edge probability $p$.

**Large Scale Graph Generation [R2, R3]**: We experimented with random graphs from stochastic block models
(SBM) to show that our model is able to generate large graphs. In particular, we generated 10 random graphs from an
SBM with 3 communities, 5000 nodes and inter/intra-community probabilities $[[0.2, 0.001, 0.002], [0.001, 0.15, 0.004],$
$[0.002, 0.004, 0.1]]$. We randomly choose 8 graphs for training and test our model on the remaining 2 graphs. The
performances of our model and several Erdos-Renyi random graph baselines are shown in Table 1. The run-time of
generating 3 graphs (5K-nodes) in parallel on a GeForce 1080Ti GPU is around 279s. We experimented with the public
implementation of GraphRNN which runs out-of-memory under this graph size. We are investigating the suggested
real-world graphs and will include the results in the final version.

**How to determine graph size [R2, R5]**: We set a maximum graph size which determines when the GNN stops. Then
we sample the graph size from the empirical distribution of graph sizes (collected from the training set). Finally, we
crop the generated adjacency matrix from top-left using this sampled graph size. Although learning to predict the graph
size is arguably better, we found this simple sampling strategy works well in practice.

**To R1**:

**- Evaluation.** We first compute the statistics, $e.g.$, degree distribution, of the generated and observed graphs respectively.
Then we measure the similarity between them using the maximum mean discrepancy (MMD) with a Gaussian kernel
using total variation distance. For spectra, $i.e.$, eigenvalue distributions, we discretize the range and treat it as a
histogram. In general, it is hard to measure differences using a single metric, as each statistic is just one particular
summarization of the graph. Low MMD scores are necessary but not sufficient for two graphs to be similar. In our
experience, small differences of MMD scores in degree distribution and spectrum, $e.g.$, 0.01 vs. 0.02, seem not to imply
a significant difference in the quality of the visualized graphs whereas clustering coefficients and orbits correlate better
with the visual quality. We will discuss this point further, and show examples along with the metrics.

**To R2**:

**- How Eq. (11) is trained?** $q(\pi|G)$ is a 2-layer GCN which takes rows in the adjacency matrix as the input node
representation. We also tried random node features and obtained similar performance. There is no ground truth for
training this network; the only training signal comes from maximizing the ELBO.

**- Maximum graph size.** Please refer to the section above. **- a in Eq.(5).** They are scalars and normalized by the
sigmoid. **- Remove RNN.** We have since done this; see above. **- Overall Training.** It is trained by teacher forcing. All
subgraphs are trained in parallel as in a pixel CNN. **- Model Size.** On the protein dataset, the size of the best-performing
GraphVAE vs. GraphRNN vs. GRAN-20 is roughly 128MB vs. 2MB vs. 4MB. We will include it in the final version.

**To R3**: Please see our response above on large scale graph generation.

**To R5**:

**- Multiple splits.** Thanks for the suggestion! We will cite that paper and add the results in the final version. **-**
**Optimization of the GNN.** We used the Adam optimizer with learning rate 1.0e-2 and decay by 0.3 at 100, 500, and
1000 epochs. The norm of the gradient is clipped at 5. We will discuss it in the final version. **- Long range dependency.**
We believe our model with the GNN and less number of generation steps do have a positive impact on this. But it is not
easy to demonstrate this explicitly through an experiment. **- Graph size.** Please refer to the graph size section above.

[Meta-Review · NeurIPS 2019]

The reviewers agree that this paper has a number of interesting ideas, and makes a useful contribution to the literature. Please take the reviewer comments into account while preparing the camera ready version. A concern raised in the discussion was about the new ablation study, showing that the RNN is not needed and training a GNN longer can achieve similar performance. This muddies the message of the paper and weakens some of the claims. This should definitely be taken into account for the final version